# Is a “Good Death” at the Time of Animal Slaughter an Essentially Contested Concept?

**DOI:** 10.3390/ani7120099

**Published:** 2017-12-14

**Authors:** Qurat ulAin, Terry L. Whiting

**Affiliations:** Manitoba Agriculture, 545 University Crescent, Winnipeg, MB R3T 5S6, Canada; Qurat.ulAin@gov.mb.ca

**Keywords:** Shechita, Dhabīḥah, essentially contested concept, animal ethics, veterinary ethics, humane slaughter, religious freedom, food ethics

## Abstract

**Simple Summary:**

The question of how to kill animals for food has persisted unresolved in the Anglo-American and European social and political discourse for more than a century. Scientific informed narrative has been directed at “documenting” the experience of the slaughtered animal in the last few seconds of life. Other narratives include wide social informed narratives of cultural, historical and religious meanings of food. Slaughter by rapid exsanguination is examined as an “essentially contested” concept as a response to the resiliency of this question in modern society.

**Abstract:**

The phrase “essentially contested concept” (ECC) entered the academic literature in 1956 in an attempt to better characterize certain contentious concepts of political theory. Commonly identified examples of contested concepts are morality, religion, democracy, science, nature, philosophy, and certain types of creative products such as the novel and art. The structure proposed to identify an ECC has proven useful in a wide variety of deliberative discourse in the social, political, and religious arenas where seemingly intractable but productive debates are found. Where a strongly held moral position is contradicted by law, a portion of the citizenry see the law as illegitimate and do not feel compelled to respect it. This paper will attempt to apply the analytic structure of ECC to the concept of animal wellbeing at the time of slaughter specifically a “good death.” The results of this analysis supports an understanding that the current slaughter debate is a disagreement in moral belief and normative moral theory. The parties to the dispute have differing visions of the “good.” The method of slaughter is not an essentially contested concept where further discourse is likely to result in a negotiated resolution. The position statements of veterinary organizations are used as an example of current discourse.

*Our concerns have nothing to do with the expression of religious beliefs but with the practice of killing by throat-cutting without pre-stunning.*
British Veterinary Medical Association 2015 [1]

## 1. Introduction

In January 2016 the Canadian Veterinary Medical Association (CVMA) adopted a position statement under their animal welfare advocacy initiative, which in part reads “slaughter-without-stunning is currently allowed in Canada under certain circumstances, the CVMA is opposed to the practice as it causes avoidable pain” [2]. The political discourse on secular-religious slaughter has been ongoing for more than 120 years in Anglo-American society [3,4]. Slaughter methods were first regulated in Canada in 1959 by a national law restricted to meat packing businesses that were eligible to export meat to the United States [5,6]. Enforcement of the law was further restricted to inspectors appointed, designated, or employed under the Meat Inspection Act (Canada). The 1959 instrument has been collapsed into the Sections 61 to 80 of the current Meat Inspection Regulations [7]. The veterinary profession has historically been core to the regulatory infrastructure assuring the welfare of animals at the time of slaughter. The veterinary profession in Europe, the United States, and Canada has claimed authority in the animal welfare debate [8]. 

For the purposes of this paper, we read from the position statement that the CVMA may support regulatory prohibition of slaughter without stunning, that is, promote the use of legal force as a tool to assure a specific conceptualization of a “good death” at the time of slaughter. The social construct of a good death for food animals is unrelated to the voluminous literature on the concept of an end-of-life good death in human animals [9,10]. This commentary will use the political frame of an essentially contested concept to focus on the conceptualization of a “good death” at the time of slaughter; specifically, the broader concerns of the slaughter-without-stunning question, as reflected in current veterinary organizational positions.

## 2. Essentially Contested Concept 

The ECC is part of a greater discussion of how society can choose to put democracy in action. Ideal deliberative democracy is a form of democracy in which open public deliberation is central to decision-making. In practice, political debate adopts elements of individual freedom, consensus decision-making and majority rule. In deliberative democracy authentic deliberation, not mere voting, is the primary source of legitimacy for policy decisions and the law [11,12]. The practice of discursive democracy holds that public arguments must be equally accessible to all persons especially when the debate is related to the government use of coercive force or fair distribution of public goods. Similarly, justification for political decisions must be formulated in language equally accessible to all citizens. 

There are genuine social and political disputes that, over a sustained period of honest effort, remain unresolved and are sustained by rational arguments and factual evidence on both sides [13]. For example, current efforts are being made on defining and achieving international agreement on the concept of “human dignity” [14]. Remarkably, in the absence of a clear mutually understood definition, “human dignity” has proliferated in wording of international agreements related to human rights [15]. 

The phrase “essentially contested concept” (ECC) entered the academic literature in 1956 in an attempt to better characterize certain contentious concepts of political theory [16]. Commonly identified examples of contested concepts are morality, religion, democracy, science, nature, philosophy, and certain types of creative products such as the novel and art [17]. The structure proposed to identify an ECC has proven useful in a wide variety of deliberative discourse in the social, political, and religious arenas, where seemingly intractable but productive debates are found. The failure of political discourse is to be avoided as the natural outcome is the galvanizing of positions and the application of state intervention in a situation where a portion of the citizenry see the law as illegitimate and do not feel compelled to respect it [18]. This paper will attempt to apply the analytic tool of ECC to the concept of animal wellbeing (a pseudo-scientific construct) at the time of slaughter specifically a “good death” (a moral-ethical construct) [19].

Re-framing an important issue as an ECC may allow for a more productive discussion between opposing views of the concept with the ultimate hope of developing an understanding between parties so that the issue becomes de-contested. In the political process, the recognition of something as an ECC allows space for the government to withdraw from the contested issue and to refrain from the exertion of force on the minority to conform. For example, in Canada, after significant public discussion of the citizen and societal meaning of voluntary termination of pregnancy, abortion was deregulated [20,21,22]. Achieving de-escalation of the dialogue avoids a drift to fundamentalism/totalitarianism. Modern, pluralist, liberal, and social democracies regard ongoing civil disagreement as healthy and enriching as the participatory citizens benefit from the opportunity to improve skills and understanding in deliberative democracy. Healthy civil disagreement decreases the risk of tyranny of the majority [23,24]. 

In an original description of the analytic framework, Gallie (1956) suggested there were five core characteristics and two additional less essential characteristics that jointly define an ECC [16]. In application to specific ideas for consideration as ECC’s, the seven characteristics (5 + 2), can vary in relevance and weighting depending on the specific conditions of the problem. The criteria are not all essential and do not stand alone but are interrelated to provide an explanatory framework or matrix for discussion [25,26]. The seven characteristics will be identified in this paper by the use of Roman numerals. 

The current debate on methods of slaughter is highly value laden. In any debate, language is a tool to seduce thought processes of others in particular directions. The use of some common phrases in the current slaughter debate are clearly appraisive or value loaded. Ritual-slaughter is commonly used as the descriptor in discussion of slaughter without stunning [27,28,29,30,31,32]. “Ritual” can be a pejorative term in the secular lexicon as it has connotations of, primitive, ignorant, meaningless, and destructive [33]. In an extreme example, Freud used “ritual” with a very constricted meaning essentially applied to repetitive, pathological compulsive behaviours of the neurotic [34]. Ritual, in secular usage, is also a practice of ancient, less cognitively developed peoples [35,36]. In current secular use, “rituals” can be framed as an exercise of choice behaviour (unnecessary) with trivial importance [37].

On considering if a concept is eligible to be considered an ECC, there must be fair agreement that the two camps hold conceptualizations taht are versions of the same concept and are not different concepts erroneously considered as the same. On the major moral issues related to the production of meat, the two camps are in high agreement. There is agreement in the mainstream animal used as food community, that death is not a harm to animals, as, with the exception of the carcass, there is no existence of anything of the individual animal post death (annihilation thesis) [38]. Fundamental issues, e.g., that it is appropriate to convert live animals into food for human consumption, food safety issues, food quality issues and that animals should be killed causing the least possible pain and distress (as well as food safety and food quality issues), are in agreement. Specifically, all parties agree that animals should not undergo unnecessary suffering through the slaughter process and be treated with care and respect [31]. Death caused by exsanguination is the primary goal of both religious and secular killing at the time of slaughter [39]. It appears that secular slaughter and Shechita or Dhabīḥah are very close to the same physical action, the killing of an animal by causing severance of the major blood vessels of the neck or thoracic inlet resulting in death by exsanguination. 

If the participants behave as if they were contesting the same concept than this mutual recognition is taken as evidence that there is agreement on the concept contested [26]. The good death at slaughter question may tentatively be considered an ECC if it reasonably fulfills the Gallie (1956) theoretical criteria, Table 1.

### 2.1. I. Appraisive

An ECC signifies some kind of valued achievement. Humane slaughter and animal welfare prior to slaughter are clearly the focus of opinions of what is good and what is right in determining a “good death” of a food animal. Describing a food as Halal or Kosher communicates the positive attribute that the thing is in compliance with the religious law, and is good and appropriate for people of faith to consume [40]. The term humane slaughter is quintessentially appraisive as the negative corollary is inhumane slaughter. The scientific literature, although not discussed in this paper, examines methods of slaughter in clearly an appraisive way with pain minimization given high concern. All parties to the discussion value the method of slaughter and believe that there is a good and an unacceptable or poor way for an animal to die in the production of meat [41].

### 2.2. II. Internal Complexity

Criteria II and III, complexity and describability are closely intertwined. The conversion of a live animal to meat portions is a complex multi-step process of both secular and religious preparation of food. The number of processes allows for actual differences in technique and is subject to widely divergent valuation of the importance of each step. At the commercial scale, broiler chickens can currently be killed at a rate of 13,500 per hour [42] placing a significant technical complexity challenge to assuring consistent good death at the individual animal level. 

In Halal slaughter, the human intent to kill the animal for food is an essential component as accidental death in livestock results in carrion not meat [39]. The application of Halal principles is complex in application and differs geographically in details [43]. The human intent to kill for food is not mentioned in descriptions of secular slaughter practices although it is an inherent aspect of industrial meat production. In the manifestation of religious belief, Dhabīḥah the animal is blessed with the name of God (Tasmiya) while being killed for human consumption. Prayer over food is common to many religions and cultures, for example almost 50% of American households “say grace” at the introduction of a family or community meal [44]. 

The process of animal slaughter can clearly be disaggregated into smaller units where good or poor quality assertions could be made on a unit basis and is clearly a complex concept and a complex physical action.

### 2.3. III. Diverse Describability 

To frame, is to select certain details of a perceived reality and make them more salient in a communication, written or spoken in such a way to restrict thought to a particular problem definition, casual interpretation, moral evaluation or obvious recommendation [45]. For example; a recently published scientific review on ruminants slaughtered without stunning from New Zealand is focused solely on the pain experienced by animals in the last two minutes of their lives, a privilege for the consideration of pain in the discourse of a good death [46]. In comparison, in an animal welfare audit of religious slaughter in Europe, the components audited were the restraint, stunning and slaughter methods, a much wider consideration of factors contributing to this specific European description of a good death [43]. It is possible to argue that the welfare of animals intended for slaughter, the objects of moral concern, originates at the feedlot or farm of origin, or earlier. The secular slaughter as “humane killing” tends to focus on the point between initial skin incision and loss of sensibility. The religious slaughter description reflects a far broader human-animal bond, impacts to individual identity and acting in compliance with canonical texts in relation to the holy food law. Both perspectives emphasize humane care and handling prior to slaughter. 

### 2.4. IV. Openness

ECCs are “open” in that as new information or sensibilities develop, the concept can be revisited and refined by both contenders. Competitors vying for hegemony over the meaning of the concept must recognize, represent and account for new information or new circumstances when defending their version of the good. In the slaughter discourse relevant information is put forth from a range of sensibilities from agnostic science informed to faith and tradition informed. Openness has an integral component of honesty. 

In considering the British veterinary statement on method of slaughter, it is dubious the British Veterinary Medical Association can credibly state that their “concerns” have nothing to do with the expression of religious beliefs. Common reasoning suggests there would be no concerns in the absence of the “Others” expression of religious beliefs [4,28]. Denial of the existence of a thing is not a satisfactory method to recognize, represent and account for weighting in a fair and open debate.

### 2.5. V. Reciprocal Recognition

This criterion presumes that the contending parties are aware that the concept under discussion is being contested by the other parties. The nature and details of contending arguments are also often bilaterally known. When one group is in the political process of trying to regulate the behaviour of another, there is little opportunity to be ignorant of the conflict. Shechita was contested long before there was law to encourage animal welfare at the time of secular slaughter [47,48]. In both Judaic and Islamic teaching, care of animals has been a significant issue and point of religious practice for many generations of the faithful [49,50].

In the public discourse of religious and cultural controversies, some groups may be significantly handicapped. Religion- based political parties are not recognized in secular democracies. In the recent extensive debate in the Netherlands on the proposal to prohibit slaughter by exsanguination, religious groups were largely excluded from the political debate [31]. While there is a registered single issue political party for animals “Partij voor de Dieren” there cannot be a “Partij voor de Jood” in a liberal or social democracy resulting in a “Dialogue of the deaf” in relation to religious meanings of animals [51]. In the Netherlands debate (faith-based derogation from compulsory pre-slaughter stunning), only two fundamental principles were given weight. Food choice was recognized as a “real” expression of individual identity and a “real” expression of group solidarity [31]. Individual identity and group solidarity are moral goods recognized as legitimate by the state. The frame of the question in the ND political debate did not extend to the possibility of slaughter being a core religious practice, nor was religious expression a valid concern in the slaughter debate. The framing of the discourse in The Netherlands denied the possibility that food can have a deeply religious meaning in the life as experienced by individuals [37]. In this particular slaughter debate, there was no mutual recognition of equal validity of the other arguments. It is common for the state to retain the agenda and decide what issues are “religious” and what issues are secular [52]. 

### 2.6. VI. Original Exemplar

Gallie described an original exemplar as the example or authority that is recognized by all vested participants [13,16,26]. The original exemplar anchors the contested concept in a historical or recognized philosophical context and assures the issue is not a simple matter of confusion. Kekes (1977) questions why is it appropriate to assert that the old ways, are better than the current ways of doing things (giving a privileged position to history) and generates an argument that this criterion is not necessary [53]. In the case of slaughter however, it may be useful to consider an imaginary perfect archetype or exemplar of “good death” from the various current moral conceptualizations. 

In the self-reflection of an ethical carnivore, moose meat (*Alces alces*) may be the least problematic of the animal muscle available for human consumption [54,55]. Sweden has large areas of remarkably productive moose habitat intermixed with sustainable forest production [56]. Legal and sustainable moose hunting for commercial moose meat provides the Swedish ethical carnivore the option to purchase wild source moose meat [56,57]. Imagine a young moose that has never been exposed to veterinary drugs, never been threatened by a human being, and never confined in an enclosure, is fully living in the state of nature, and has never been lied to [58]. Standing in a bog, this young moose suddenly absorbs 2500 J of energy (common high powered rifle load) with his skull and brain, resulting in immediate painless death. This possible moose scenario could be considered the quintessential secular good death within the eco-humane frame of providing meat for human consumption. However, the rare possibility of immediate insensibility and the high animal welfare risk of injury and escape, given the technical realities of hunting, place the practice of hunting itself in moral question [59,60]. 

The Binding of Isaac is a story from the sacred text of Christian, Islam, and Jewish faith found in Genesis 22. In the Genesis KJV narrative, in order to test Abraham’s faith, Abraham, the archetypal man of faith (in Islam, a Prophet) is requested by Yahweh to sacrifice his son Isaac as a burnt offering, on Mount Moriah. The Christian/Judaic narrative focuses primarily on the spiritual experience of Abraham, the attitudes of the participants, the action of deception of Isaac, and, as necessary for the story, the physical activities [61,62]. The Qur’anic version is significantly different and communicates a more comprehensive statement of faith. Ibrahim tells his son (Ismael) of his dream of the religious sacrifice of Ismael. Ismael responds: if it is the will of God then the path for us is clear. This story also embodies faith but more importantly it demonstrates that faith includes humility to recognize the limits to human knowledge, even the knowledge of a Prophet and a complete human life includes faith [63]. The narrative may parallel the idealized practice of religious slaughter: a component of which is obedience to the will of the divine.

The oral history of the Algonquin first nations (Eastern Canada) places hunting in a religious frame [64]. The belief in supernatural forces and the existence of a spiritual connection between the hunter and the hunted creates a human-animal bond where the prey arrives and of its own free will gives their life to the hunter, making the sacrificial death of the animal an act of forgiveness and negation of guilt [65]. When animals are represented as other-than-human persons, who give themselves to hunters as part of a continuous human-animal relationship of reciprocal exchange, animal pain involved in the process of dying in the hunt is not part of the narrative [66]. 

### 2.7. VII. Progressive Competition

Early work on the notion of ECCs was motivated by the expectation that working through the process of a standard approach to these kinds of issues would increase the quality of the arguments in the interaction of the contested parties. In the best of worlds, previously contested issues could, over time, become de-contested [67,68]. 

## 3. Results and Discussion

The concept of a good death at the time of slaughter does not fit well in the mold of an EEC. In applying the ECC template to the concept of slaughter it fails Criteria IV openness; similar arguments founded on religious belief are not considered accessible to people without faith and are thereby excluded from the public discourse. For proponents of slaughter by exsanguination, opinions from physiology are not compelling. Both sides in the debate appear “closed” and rational consideration of the opposing view is not possible.

Criteria V, reciprocal recognition, also fails as not all sorts of information is recognized by all parties to the conflict. The secular argument in essence does not recognize the religious argument as “rational” and therefore the religious perspective is invalid. There is no agreement on an exemplar (Criteria VI), and there does not appear that the discourse is progressing towards a mutually satisfactory solution, Criteria VII. 

The conceptualization of good death as articulated by secular-scientific proponents via pre-slaughter stunning is a different conceptualization than the “good death” recognized by faith communities. Most dramatically, secular slaughter is an action involving a non-human animal and a human animal, whereas religious slaughter is an action involving a non-human animal, a human animal and the divine (Dhabīḥah) or the divine law (Shechita) [69]. In secular slaughter, the animal is at the center of the discourse. In religious slaughter, most consistently in Halal slaughter, God is at the center of the discourse. The method of the slaughter debate is a disagreement in moral belief and normative moral theory [70]; the parties to the dispute have differing visions of the “good.” It appears from this examination that the social conflict related to slaughter by exsanguination is not an ECC and continual arguments from the current paradigms will not converge toward a consensus.

The slaughter debate is frequently portrayed as a difference between secular society backed by science verses faith traditions founded on divine revelation and centuries of experiential value [23,71]. Spiritual quality is put forward as an essential food attribute in context of food grading for the religious consumer [40]. The parameter of ‘spiritual quality,’ like the organically cannot be measured post production as other attributes like color, consistency, flavor, or juiciness of meat. Spiritual quality is a fundamental quality component to those who hold it to be of value, the absence of which renders the meat spiritually worthless and not consumable [40,72]. 

### 3.1. Veterinary Special Interest

National veterinary associations in Anglo-American and European countries have almost unanimously expressed the opinion that slaughter without stunning is a serious welfare concern; United States is an exception where the national veterinary organization is largely silent [73]. The Federation of Veterinarians of Europe have expressed: “from an animal welfare point of view, and out of respect for an animal as a sentient being, the practice of slaughtering animals without prior stunning is unacceptable under any circumstances” [74]. This is an extreme and unequivocal condemnation of a millennial old practice. This almost singular voice of first world veterinary organizations begs for an explanation. 

Professions and the image of professions are largely created, maintained and defended by members of the profession themselves [75,76]. The authors pose that the method of slaughter issue cuts to the core of the current narrative myth of the Anglo-European veterinary profession in both philosophical and practical ways. In the past half century, the veterinary profession in the EU and other Western democracies has been politically re-created as the a priori animal welfare advocate [77,78]. The decision to euthanize companion animals is almost always subsequent to a deep professional-client discussion of significant personal impact [79]. The human-animal bond and the utilization of emotional intelligence are now part of the cultural indoctrination of students [80]. In companion animal practice animal welfare and pain control have relatively recently become common or the standard of practice [81,82,83]. Animal pain initiates a different dialogue within the veterinary profession now as compared to 25 years previous. 

Veterinary professional organizations oppose religious slaughter with the motivation of preventing pain. The avoidance of animal pain is not always given a privileged position in other areas of veterinary influence. For example, bilateral international phytosanitary agreements are negotiated by the national veterinary infrastructure of the two countries. International health certificates for live animal trade are entirely in the control of the veterinary profession. Canada and the United States have agreed that hot-iron branding three symbols (C∧N, C lambda N) not less than 2 inches in height [84], is an acceptable animal identifier for the purposes of live animal trade in cattle [85], no exemption exists for dairy calves less than 10 days old. Hot iron branding is internationally recognized as a form of torture when applied to human animals [86,87]. Similarly, although firefighting foam kills poultry by suffocation [88], it is approved in foreign animal disease eradication [89].

This significant question of the cause of disproportional concern of the veterinary profession in methods of slaughter is beyond the scope of the commentary and deserves specific attention by sociologist and organizational experts. 

### 3.2. Canadian Considerations

In Canada, there is no national regulatory instrument to guide the methods of slaughter in facilities not authorized to participate in international trade. The Canadian constitution places the primary responsibility for social programs and the realm of commerce with the provinces while criminal justice remains with the federal authority [90]. There has been little progress in the multi-decade social movement to modernize the animal protection section of the Federal Criminal Code (Canada), which has remained largely unchanged for 120 years [91,92,93]. The Code also retains a provision for conviction for blasphemy [94], reflecting a tradition of church-state overlap in Canada. In the absence of comprehensive national public support, there is a reluctance for the Canadian Government to regulate in areas outside health care and education [95]. For example, there is no national policy on child care despite decades of social activism [96]. 

Secularization is a process initiated in the Enlightenment as a response to the violent European civil wars of religion, which waged from 1524 (German Peasants War) to 1648 (end of the Three Kingdoms War). The separation of religion and government (separation of church and state) was one aspect of the Enlightenment movement. The Enlightenment movement dominated the world of ideas in Europe during the 18th century and continues as the basis of Western democracies. Over this time, religion was progressively marginalized to where it no longer functions as an explanation of the world, no longer facilitates social cohesion or personal identity and takes on the status of a choice or hobby [97]. The Enlightenment hubris endorses the preeminence of science over all other ways of knowing and science pre-supposes a single truth, not a range of alternate truth which is manifest in multi-denominational expressions of faith [98]. Pure separation of religion and state would require both no government support for religion and no government interference with religion. In a comprehensive study of 154 national governments of countries with over 1 million citizens, only the United States was identified as having complete separation of religion and government [99]. 

Secularism can manifest as a very intolerant ideology [98,99,100]. One manifestation of hard secularism is the implementation of laïcité in France. Laïcité is the legal requirement placed upon French civil servants and organizations to observe religious neutrality in act, deed, and appearance. The government census in France and many European countries cannot ask for information related to religious practices [101], as religiosity is officially outside the concern of government. In laïcité application, religious observance and visible symbols of faith must be restricted to the privacy of the home, such as the prohibition of traditional dress choices for Muslim women in publically funded schools [102,103,104]. More comprehensive authoritarian forms of secularism include the official policy of the historic USSR [105]. Authoritarian states enforced secularism is currently practiced in Singapore [106]; with Turkey [107] and Egypt [108] practicing a more mixed approach of significant government participation in accepted religious practices.

Canadian society as a whole manifests an “open secularism” [109]. The Canadian public generally supports the concept that individuals of faith, hold faith as part of an integrated individual identity and should not be prohibited from publically communicating that faith identity to others [110,111]. Acceptance of personal faith markers is manifest in the modification of the RCMP uniform to replace the Stetson hat with the Sikh turban when desired by the affected individual [112]. The Canadian Charter of Rights and Freedoms as part of the Canadian Constitution has been characterized as a “Muscular form of Liberalism” [113], providing significant constitutional protection for religious freedom. The Canadian Constitution has also retained a reference to God as a basis of authority of government [114]. Multiculturalism requires the development of new civic and political relations to overcome the deeply entrenched inequalities that have persisted since the abolition of formal discrimination [115]. 

A post-Enlightenment citizenry should recognize that, when considering coercive policies of the state, there are a certain “kind of reasons” that cannot be ignored, disregarded, or overridden [111]. Deeply held religious convictions, that play an integral role in the life of a person of faith, reach this special status of reasons that cannot be ignored. In a Canadian political debate of the humaneness of hunting, the Algonquin hunting exemplar given here, may exemplify one of these “kind of reasons” that preclude regulation of first nations hunting in the context of post-colonial Canadian nation [113]. Canada and Canadians continue to recognize the current manifestations of poor treatment of aboriginal peoples under the colonial and post-colonial racist policy [106,116]. 

In Canada, to date, the method of slaughter debate has not generated significant public interest. Original national regulation of the issue was not initiated as an animal welfare concern but, to facilitate international trade. There is no clear consensus among the general Canadian public on the method of slaughter or that lack of specific regulation is a problem. 

The German Green Party wrestled over the question of slaughter by exsanguination as a religious expression for over 25 years and resolved that on the balance of social good, the risk of avoidable pain in animals did not trump all other social and individual rights concerns [117]. It is highly unlikely that Canada, a country with no comprehensive national method of slaughter law, could make the step to prohibiting religious slaughter practices.

A prominent Canadian academic (Lori G. Beaman, University of Ottawa) has suggested that the hidden attitudes represented by the words “tolerance” and “accommodation” are insufficient concepts to deal with the social pressures resulting from the current involuntary migration of peoples and the commitment to justice required for the existence of democracy. Tolerance and accommodation are things the dominant culture can bestow, from its abundance of generosity, upon the “Other” who is in no position to do other than accept with humble supplication. This is not compatible with current conceptualizations of justice [118]. The concept of “Deep Equality” gives hope to the future of a more communal and less violent world [119,120].

## 4. Conclusions

This commentary has significant limitations. The ECC as originally designed was considered a thought experiment, as a rhetorical tool and not an application to the real external world. It was intended as an aid to the logical, epistemological, and psychological difficulties in making sense of the real world. Authors using this model commonly acknowledge that it was not Gallie’s intent for his logistic model to be applied to real world situations, and then go on to ignore that intent and apply it to real world problems. There may be better tools for understanding the moral positions related to livestock slaughter than the EEC construct. Although the attempt to frame the method of slaughter debate as an ECC has appeared to fail, the process has identified components of the argument where open discussion may improve the quality of the public discourse. Better individual understanding may start with a reasonable uncertainty replacing a conviction of infallibility whenever possible. 

The method-of-slaughter construct as a particular social, ethical, and cultural issue may require the veterinary profession to demonstrate true openness to the “Other.” True “recognition” in a society committed to multi-culturalism is the cognitive assertion, where acceptance is difficult, that the opinion of the “Other” is as valid as is mine, although we may disagree to the depth of our personal identity. Openness to the Other “involves recognizing that I myself must accept some things that are against me, even though no one else forces me to do so” [121]. 

Further studies on the evidence of pain at the time of exsanguination is unlikely to alter the trajectory of the public discourse on what is a “good death” at the time of slaughter. We predict the scientific belief that all things can be known will continue to fuel research on the moment of death experience of slaughtered animals. We suggest a recognition that this belief of attainment of perfect knowledge is an error and we cannot decrease the uncertainty around the death experience of food production animals. 

Animal welfare is a continuing evolutionary concept [49]. The authors suggest that the current position statement of the CVMA on slaughter without stunning be reconsidered in light of the Canadian history and our commitment to the future. It is unclear if the phrase “opposed to the practice” includes support for the use of state power to eliminate slaughter without prior stunning. This ambiguity should be corrected in the next revision of this position statement.

## Figures and Tables

**Table 1 animals-07-00099-t001:** Characteristics of an essentially contested concept (ECC).

Characteristic	Description
I. Appraisive	It signifies or accredits some kind of valued achievement, considered an important social-moral-political subject of public concern
II. Internal complexity	The contested concept is not simple but constructed of smaller ideas
III. Diverse describability	Different positions of the concept will rank the worth of those internal parts of the concept differently
IV. Openness	Must be of a kind of concept that admits considerable modification in the light of changing circumstances; and such modification cannot be prescribed or predicted in advance
V. Reciprocal recognition	Different users of the concept must reciprocally recognize its contested character among contending parties by using the concept both aggressively (against other conceptions) and defensively
VI. Original Exemplar	The presence of a real or imaginable archetypal example to support the understanding of the concept
VII. Progressive competition	The act of continual discourse, effective competition about the usages of the concept results in a greater coherence of the concept for all participants

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
