# Peer review of "Is a “Good Death” at the Time of Animal Slaughter an Essentially Contested Concept?"

_animals, 2017, doi:10.3390/ani7120099_

Round 1

Reviewer 1 Report

This is a well-reasoned manuscript examining the underlying philosophical and moral concerns surrounding recent attempts by governments and professional veterinary organizations to influence and legislate Halal slaughter practices as serious breaches of animal welfare. Basically, the authors question whether the state should be using its legal power to force compliance on what is essentially a morally held religious position.

The issue of animal welfare associated with halal slaughter has the superficial appearance of an “essentially contested concept” (ECC) as originally defined by Gallie, where the process of working through seemingly intractably contested positions taken by the conflicting parties through public discourse eventually leads to resolution and understanding such that the issue is decontested. The authors review the multiple criteria which define an ECC in light of and with specific reference to the process of turning animals into food. The authors conclude that the concept of a “good death” at the time of slaughter does not fit into the concept of an ECC, as arguments based on religious belief are not considered accessible to people without faith and physiology-based arguments are not compelling to people of faith. Both sides in the debate are therefore considered “closed” with rational consideration of the opposing view not possible. As such, the issue likely cannot be resolved.

The authors extend their discussion to formal statements made by national veterinary associations in North America and Europe on slaughter without stunning as a means to prevent animal pain, while citing several inconsistencies where similar regard to animal pain is ignored by these same organizations. Further discussion examines whether the use of state force, through legal fiat, is justified on this issue; a secular legal resolution will be coercive and a disadvantage to the religious believer.

The manuscript is timely, thought provoking, and will generate substantial discussion by the profession.

Author Response

Reviewer 2 animals-241742 “Good death at the time of slaughter”

This is a well-reasoned manuscript examining the underlying philosophical and moral concerns surrounding recent attempts by governments and professional veterinary organizations to influence and legislate Halal slaughter practices as serious breaches of animal welfare. Basically, the authors question whether the state should be using its legal power to force compliance on what is essentially a morally held religious position.

The issue of animal welfare associated with halal slaughter has the superficial appearance of an “essentially contested concept” (ECC) as originally defined by Gallie, where the process of working through seemingly intractably contested positions taken by the conflicting parties through public discourse eventually leads to resolution and understanding such that the issue is decontested. The authors review the multiple criteria which define an ECC in light of and with specific reference to the process of turning animals into food. The authors conclude that the concept of a “good death” at the time of slaughter does not fit into the concept of an ECC, as arguments based on religious belief are not considered accessible to people without faith and physiology-based arguments are not compelling to people of faith. Both sides in the debate are therefore considered “closed” with rational consideration of the opposing view not possible. As such, the issue likely cannot be resolved.

The authors extend their discussion to formal statements made by national veterinary associations in North America and Europe on slaughter without stunning as a means to prevent animal pain, while citing several inconsistencies where similar regard to animal pain is ignored by these same organizations. Further discussion examines whether the use of state force, through legal fiat, is justified on this issue; a secular legal resolution will be coercive and a disadvantage to the religious believer.

The manuscript is timely, thought provoking, and will generate substantial discussion by the profession.

 Reply:

We thank you sincerely for the time and effort you have contributed in reviewing this project. Both authors are from the veterinary community and we have increasing concerns with the state of mind of the profession in Europe and Canada in relation to this issue. We are very pleased that there may be an audience in the profession for the message of this manuscript.

Sincerely;

The authors 

Reviewer 2 Report

Authors of this paper tried to use the method of ECC to explore the solution of "To get rid of a deliberate animal humane slaughtering method (OIE Terrestrial Animal Health Code Chapter 7.5) for non-reasoning traditions or beliefs."  This paper initiates a very equivalent and comprehensive way to assess a moral issue. It is firstly in the world to use this method to assess the morality of slaughter of animals.  

For anyone having the ability to imagine oneself in the position of another entity (human or nonhuman), will always feel a moral duty to present the issue of this paper to the public. The plain truth is that many of some cruel customs and traditions have died out, such as: slavery, Roman gladiatorial contests, torture, public executions, witches burning, racism, etc. This paper initiates a realistic way of justice for nonhuman animals --- People owe animals justice. The reason is simple: nobody among those who insist slaughtering without stunning still insist the medication way to treat leprosy written in the Pentateuch. The fact points out their contradiction of the perception of their beliefs/religions/customs/traditions.   

Author Response

Reviewer 1

Comments and Suggestions for Authors

Authors of this paper tried to use the method of ECC to explore the solution of "To get rid of a deliberate animal humane slaughtering method (OIE Terrestrial Animal Health Code Chapter 7.5) for non-reasoning traditions or beliefs."  This paper initiates a very equivalent and comprehensive way to assess a moral issue. It is firstly in the world to use this method to assess the morality of slaughter of animals.  

For anyone having the ability to imagine oneself in the position of another entity (human or nonhuman), will always feel a moral duty to present the issue of this paper to the public. The plain truth is that many of some cruel customs and traditions have died out, such as: slavery, Roman gladiatorial contests, torture, public executions, witches burning, racism, etc. This paper initiates a realistic way of justice for nonhuman animals --- People owe animals justice. The reason is simple: nobody among those who insist slaughtering without stunning still insist the medication way to treat leprosy written in the Pentateuch. The fact points out their contradiction of the perception of their beliefs/religions/customs/traditions.   

Reply

We thank you sincerely for the time and effort you have contributed in reviewing this project.

We have included reference to Chapter 7.5 in our current draft.

We are thankful for your consideration of this manuscript.

Sincerely

The Authors

Reviewer 3 Report

This paper is an extremely well-documented and thought out presentation of a topic the authors are clearly passionate about. In particular the second half was very well stated. I therefore have some suggestions on things to shorten in order to keep the reader's attention so that he gets to the main points. The use of ECC as a model was a strawman to arrive at the second half after rejecting the utility of ECC in this debate. Thus, I recommend shortening section 2 which is a long, too long, introduction to this concept that admittedly I was not familiar with prior to reading this.

In section 2.3 the authors note the fixation on the final 2 minutes of the animal’s’ life. This is a very important point and can be highlighted by noting the huge sums invested by the EU in research on the topic. Most recently the 6 figure BOREST grant focused solely on restraint of bovine in those last two minutes. While any suffering in that period should be minimized, the authors’ point that that is not the sole issue is important. This point was again made in the important last paragraph in 3.1. Although you beg off elaborating, if you have a theory I think it should be mentioned.

In 2.5 it is noted that Shechita was contested long before … . It might be worth noting if the authors feel anti-Semitism (and in today’s Europe anti-Semitism and Islamophobia) play any role in this debate. And noting the Judaic laws encouraging animal welfare long before Western states had such laws.

Within 2.6 I do not see the contribution of the paragraph on the Binding of Isaac. (In addition you lump together the three Abrahamic religions – regarding this story Islam has modified it to their needs and cannot be lumped like this.)

Section 3.1 is interesting in analyzing how a profession coalesces around a position, in this cases veterinarians against religious slaughter. Similar analysis could be done regarding pediatricians and circumcision. Or, and I think it should be mentioned in this section, how veterinarian groups have come to in general support neutering, a practice that would seem to be problematic from an animal welfare perspective and which more and more evidence shows can be medically detrimental. But you should be careful about accusations of being disingenuous as I suspect that work is being done to minimize other painful procedures as well.

I realize the authors are Canadian, but I think the section on Canada is too long and should be shortened, again so as not to lose the reader.

Finally, despite the last paragraph, I suspect that both sides in this debate will continue with scientific studies in attempts to sway the other side and legislators.

Author Response

Reviewer 3

This paper is an extremely well-documented and thought out presentation of a topic the authors are clearly passionate about. In particular the second half was very well stated. I therefore have some suggestions on things to shorten in order to keep the reader's attention so that he gets to the main points. The use of ECC as a model was a strawman to arrive at the second half after rejecting the utility of ECC in this debate. Thus, I recommend shortening section 2 which is a long, too long, introduction to this concept that admittedly I was not familiar with prior to reading this.

We recognize that all writings can be improved by a good editor. We have eliminated the paragraphs original lines 95-110 as discussing minor ideas not essential to the overall narrative.  

In section 2.3 the authors note the fixation on the final 2 minutes of the animal’s’ life. This is a very important point and can be highlighted by noting the huge sums invested by the EU in research on the topic. Most recently the 6 figure BOREST grant focused solely on restraint of bovine in those last two minutes. While any suffering in that period should be minimized, the authors’ point that that is not the sole issue is important. This point was again made in the important last paragraph in 3.1. Although you beg off elaborating, if you have a theory I think it should be mentioned.

We have addressed this in a concept way not in the original submission. One primary concept that we have inserted is the fact that “Science” by its, nature is a narcissistic, self-aggrandizing, that acts with th hubris which admits no limit to knowledge and no humility.  In contrast, Muslim faith in particular articulates that human knowledge is limited especially in comparison to the one who was not created. We try to weave his idea into the Binding of Isaac exemplar which binds this story closer to the main ideas of the paper.

In 2.5 it is noted that Shechita was contested long before … . It might be worth noting if the authors feel anti-Semitism (and in today’s Europe anti-Semitism and Islamophobia) play any role in this debate. And noting the Judaic laws encouraging animal welfare long before Western states had such laws.

We agree with this criticism and will include a statement to this effect and cite the new references(1, 2).

(and in today’s Europe anti-Semitism and Islamophobia)

We had not originally considered commenting on this; however, with your support it now seems appropriate.  We believe that anti-Semitism and Islamophobia are insufficiently broad concepts to capture what is the state of current affairs. Individual members of the white dominant culture can avoid accountability for their state of privilege by asserting AS&IP are acting only by rare subgroups in the dominant culture.  Dominant culture members can self-delude that they are not guilty in accepting the benefits that a system of white supremacy exists in Euro-American culture. One of the authors and spouse are recently immigrant to Canada and experiencing the barriers in real time.      

We are targeting this concern with new Paragraph at line 301- 307 Section 3.1 We use the term “white supremacy” with intent, hoping that the veterinary readership may be jarred into reconsidering the various position statements.

This is an extreme and unequivocal condemnation of a millennial old practice. This almost singular voice of first world veterinary organizations begs for an explanation.

The articulation of veterinary organization on the welfare of animals at the time of slaughter have intensified and clarified only in the recent past. The increased prominence of this issue at this time cannot be separated from contemporary world events. The wave of anti-immigrant rhetoric in part manifest by the exit of Britain from the European Union by populist vote(3) and the controversial election of a far right candidate in the American Presidential election described as a manifestation of racist nativism reflecting a previously disenfranchised politics of white supremacy(4).

Professions and the image of professions are created, maintained and defended by members of the profession themselves [71, 72].

Within 2.6 I do not see the contribution of the paragraph on the Binding of Isaac. (In addition you lump together the three Abrahamic religions – regarding this story Islam has modified it to their needs and cannot be lumped like this.)

We agree with this statement and propose the following adjustment.

The Binding of Isaac is a story from the sacred text of Christian, Islam, and Jewish faith found in Genesis 22. In the Genesis KJV narrative, in order to test Abraham’s faith, Abraham, the archetypal man of faith (in Islam, a Prophet) is requested by Yahweh to sacrifice his son Isaac as a burnt offering, on Mount Moriah. The Christian/Judaic narrative focuses primarily on the spiritual experience of Abraham, the attitudes of the participants, the action of deception of Isaac and as necessary for the story, the physical activities [63, 64]. The Qur’anic version is significantly different and communicates a more comprehensive statement of faith.  Ibrahim tells his son (Ismael) of his dream of the religious sacrifice of Ismael.  Ismael responds with; if it is the will of God then the path for us is clear. This story also embodies faith but more importantly, it demonstrates that faith includes humility to recognize the limits to human knowledge; even the knowledge of a Prophet and a complete human life includes faith [65].  The narrative may parallel the idealized practice of religious slaughter: a component of which is obedience to the will of the divine.

It is possible we are overreaching with the inclusion of this paragraph, however; we would like to keep it as the humility of faith assertion supports our overall message of anti-extremism especially secular-science extremism which does not get enough visibility in the scientific literature

Section 3.1 is interesting in analyzing how a profession coalesces around a position, in this cases veterinarians against religious slaughter. Similar analysis could be done regarding pediatricians and circumcision. Or, and I think it should be mentioned in this section, how veterinarian groups have come to in general support neutering, a practice that would seem to be problematic from an animal welfare perspective and which more and more evidence shows can be medically detrimental. But you should be careful about accusations of being disingenuous as I suspect that work is being done to minimize other painful procedures as well.

Accusations of being disingenuous We appreciate your identification of this issue.  We have removed it from the paper.

 I realize the authors are Canadian, but I think the section on Canada is too long and should be shortened, again so as not to lose the reader.

 Agreed: We managed to decrease the overall paper by about 500 words and eliminated about 10 original references, but, 4 new references made it in

 to accommodate the changes in messaging. So only a moderate success in this request.

Finally, despite the last paragraph, I suspect that both sides in this debate will continue with scientific studies in attempts to sway the other side and legislators.

 Agreed: We also believe that the nature of science will prohibit the acceptance to the limits of human knowledge which in integral to the discussion of this issue. Line 447-442 is also a new paragraph in response to your comment.

1.         Pozzi P, Geraisy W, Barakeh S, Azaran M. Principles of Jewish and Islamic Slaughter with Respect to OIE (World Organization for Animal Health) Recommendations. Israel Journal of Veterinary Medicine. 2015;70:3.

2.         Aidaros H. Drivers for animal welfare policies in the Middle East. Rev sci tech Off int Epiz, . 2014;33(1):85-9.

3.         Hobolt SB. The Brexit vote: a divided nation, a divided continent. Journal of European Public Policy. 2016;23(9):1259-77.

4.         Huber LP. Make America Great again: Donald Trump, Racist Nativism and the Virulent Adherence to White Supremecy Amid US Demographic Change. Charleston L Rev. 2016;10:215-48.

We sincerely appreciate your careful reading of this manuscript and are in you debt for the improvements you have suggested.

The Authors.